

# Foraging behaviour of an egg parasitoid exploiting plant volatiles induced by pentatomids: the role of adaxial and abaxial leaf surfaces

Francesca Frati[1], Antonino Cusumano[2], Eric Conti[1], Stefano Colazza[3], Ezio Peri[3], Salvatore Guarino[3], Letizia Martorana[3], Roberto Romani[1] and Gianandrea Salerno[1]

[1] Department of Agricultural, Food and Environmental Sciences, University of Perugia, Perugia, Italy
[2] Department of Entomology, Wageningen Agricultural University, Wageningen, Netherlands
[3] Department of Agricultural and Forest Sciences, University of Palermo, Palermo, Italy

## ABSTRACT

Several phases of herbivorous insect attack including feeding and oviposition are known to induce plant defenses. Plants emit volatiles induced by herbivores to recruit insect parasitoids as an indirect defense strategy. So far, volatiles induced by herbivore walking and their putative role in the foraging behavior of egg parasitoids have not been investigated. In this paper we studied the response of the egg parasitoid *Trissolcus basalis* toward volatiles emitted by *Vicia faba* plants as consequence of the walking activity of the host *Nezara viridula*. Olfactometer bioassays were carried out to evaluate wasp responses to plants in which the abaxial or the adaxial surfaces were subjected to walking or/and oviposition. Results showed that host female walking on the abaxial but not on the adaxial surface caused a repellence effect in *T. basalis* 24 h after plant treatment. The emission of active volatiles also occurred when the leaf was turned upside-down, indicating a specificity of stress localization. This specificity was supported by the results, which showed that oviposition combined with feeding elicit the induction of plant volatiles, attracting the parasitoid, when the attack occurred on the abaxial surface. Analyses of plant volatile blends showed significant differences between the treatments.

# INTRODUCTION

Plants belong to complex communities interacting with different organisms (*Dicke, Van Loon & Soler, 2009*). In particular, plants are continuously under attack from herbivorous insects since they are used as food source, oviposition site and place to meet potential mates. After the attack, plants can activate specific responses that result in defenses against the threats (*Karban & Baldwin, 1997*; *Schoonhoven, Van Loon & Dicke, 2005*; *Howe & Jander, 2008*; *Schaller, 2008*). These defenses can be direct, affecting negatively the physiology or behaviour of the herbivorous insect, and indirect, attracting herbivore enemies through the synthesis of volatile compounds, named herbivore-induced plant volatiles (HIPVs) (*Hare, 2011*; *Heil, 2014*).

Corresponding author
Francesca Frati,
francescafrati@tiscali.it

Plant attack by phytophagous insects can be divided into different phases acting in sequence or in concert (*Hilker & Meiners, 2010*). During an attack, herbivorous insects come in contact with plants by touch or walk, and then they can feed and/or oviposit on or into the plant. Plants apparently seem passive but they are capable to sense touch by wind, vibrations (*Appel & Cocroft, 2014*) and also insects (*Hilker & Meiners, 2010*). In particular, they have the perception of being touched or scratched by a walking herbivore or the capacity to respond to chemical substances released from the herbivore's tarsi during the walking activity on the plant substrate (*Hilker & Meiners, 2010*). During their foraging and/or oviposition activity, herbivorous insects leave their footprints on the plant substrate, which are an ensemble of tarsal pressure with surface and tarsal chemical secretions. Insect footprints have been investigated from different points of view. In the case of bumblebees, chemical footprints left on flowers are considered as an intraspecific signal used by conspecifics to avoid recently visited flowers (*Eltz, 2006*). Moreover, footprints have been studied considering the role played by the pad secretions in the mechanism of insect adhesion on the surface, and the quantification of the fluid secretion rate in adhesive pads (*Gorb, 2001*; *Dirks & Federle, 2011*). Finally, insect footprints can act as interspecific signal between herbivores and parasitoids. The larval parasitoid *Cotesia marginiventris* Cresson (Hymenoptera: Braconidae) can detect footprints left by the caterpillars of its host *Spodoptera frugiperda* Smith (Lepidoptera: Noctuidae) during walking activity, since the footprints are adsorbed on the plant wax surface (*Rostás et al., 2008*). In addition, it is known that footprints left on the substrate by true bugs have kairomonal effect for some *Trissolcus* and *Telenomus* species (Hymenoptera: Platygastridae) (*Colazza, Salerno & Wajnberg, 1999a*; *Borges et al., 2003*; *Conti et al., 2004*; *Salerno et al., 2006*). In particular, footprints left on the substrate by *Nezara viridula* (L.) (Heteroptera: Pentatomidae) females are perceived by the egg parasitoid *Trissolcus basalis* (Wollaston) females since they are retained onto the epicuticular waxes (*Colazza et al., 2009*; *Lo Giudice et al., 2010*; *Lo Giudice et al., 2011*). Moreover, the footprints left by *Murgantia histrionica* Hahn (Heteroptera: Pentatomidae) females on the cabbage leaves are able to induce the emission of contact synomones (*Conti et al., 2010*; *Frati, Salerno & Conti, 2013*); in fact, *M. histrionica* footprints left on leaf surface elicit a behavioural response in the egg parasitoid *Trissolcus brochymenae* (Ashmead) (*Conti et al., 2010*). In addition, these footprint-induced synomones are adsorbed by the epicuticular waxes of cabbage leaves and subsequently exploited by the parasitoid female (*Frati, Salerno & Conti, 2013*). All these examples refer to the parasitoid responses after its landing on plant. However, to date no studies have investigated whether the *N. viridula* walking activity—i.e., the ensemble of tarsal pressure with surface and the tarsal chemical secretions—is involved in eliciting the induction of specific volatile organic compounds (VOCs), which can affect the foraging behaviour of insect parasitoids. In our paper, using the tritrophic system *Vicia faba* (L.)—*Nezara viridula*—*Trissolcus basalis*, we tested whether herbivore walking induces the emission of plant volatiles eliciting a response of the egg parasitoid in olfactometer with particular attention on the role of leaf surface on which the biotic stress occurred. Behavioural experiments are conducted to test our hypothesis and they are followed by chemical analysis to confirm the recorded behavioural responses.

The walking activity on the substrate during the host plant exploitation represents the early phase of the herbivore attack. This phase can be followed by feeding and egg deposition. Plants are able to perceive herbivore feeding that is characterized by the physical damage on the plant tissue (*Hilker & Meiners, 2010*). However, the plant artificial wounding is not able to mimic the damage inflicted by herbivore feeding that is associated to the release of herbivore regurgitants into the plant wounds triggering the defensive responses (*Hilker & Meiners, 2010*). Plants can also sense insect oviposition (*Hilker & Meiners, 2006*; *Hilker & Meiners, 2010*; *Hilker & Fatouros, 2015*; *Hilker & Fatouros, 2016*). For several systems, it is well known that insect oviposition induces the emission of plant synomones that attract specific egg parasitoids (*Meiners & Hilker, 1997*; *Meiners & Hilker, 2000*; *Hilker & Meiners, 2002*; *Hilker, Rohfritsch & Meiners, 2002*; *Colazza et al., 2004*; *Colazza, McElfresh & Millar, 2004*; *Fatouros et al., 2005*; *Fatouros et al., 2007*; *Fatouros et al., 2008*; *Fatouros et al., 2009*). Generally, the oviposition-induced synomones are perceived by the parasitoids as olfactory stimuli. However, the egg parasitoids might also respond to contact synomones perceived after they have alighted on the plant (*Fatouros et al., 2005*; *Fatouros et al., 2007*; *Fatouros et al., 2009*; *Conti et al., 2010*). In our study model it is clearly demonstrated in olfactometer that *T. basalis* is attracted to OIPVs induced by oviposition combined with feeding of *N. viridula* (*Colazza et al., 2004*; *Colazza, McElfresh & Millar, 2004*). However, the feeding damage alone does not elicit parasitoid response (*Colazza et al., 2004*). Phytophagous stink bug species most frequently oviposit on the underside of soybean leaves (*Tood & Herzog, 1980*), and this behaviour guarantees to the herbivore egg masses a great protection from predators and to unfavourable microclimatic conditions (*Müller & Hilker, 2001*) due for example to sunlight. This was also confirmed for *N. viridula* on soybean plants where about the 80% of egg masses were laid on the abaxial leaf surface (*Colazza & Bin, 1995*).

On this account, do plants specifically respond to feeding and oviposition according to stress localization? In order to answer to this question, we tested in the described system (a) whether there is an indirect plant response when *N. viridula* feeding damage is inflicted on adaxial leaf surface and (b) whether indirect plant response is stronger when the oviposition (combined with feeding) occurs on abaxial part of the leaf (this situation resembles the most common case occurring in nature).

This study could give significant information regarding the ecological consequences of herbivore walking activity on plants and could contribute to understand the role of the footprints as a component of the oviposition-induced plant volatile (OIPV) induction in the studied system. In particular, this paper could give an important contribute in the knowledge of the interaction between host and egg parasitoid mediated by OIPVs.

## MATERIAL AND METHODS

### Insects

The *N. viridula* colony was reared in a controlled condition chamber (25 ± 1 °C; 70 ± 10% RH, 14 h:10 h L:D), inside clear plastic food containers (300 mm × 195 mm × 125 mm-high) with 5 cm diameter mesh-covered holes for ventilation. Separate containers were used for nymphs and adults. All stages were fed with a diet of sunflower seeds and seasonal fresh

vegetables and food was changed every 2–3 d. Egg masses were collected daily and used to maintain cultures of both *N. viridula* and the parasitoid *T. basalis*. The *N. viridula* colony was supplemented regularly with field collected bugs.

The parasitoid *T. basalis* was reared on *N. viridula* egg masses glued on paper strips. Wasps were maintained in 85 ml glass tubes, fed with a honey-water solution and kept in controlled environment room under the same rearing conditions of *N. viridula*. After emergence, male and female wasps were kept together to allow mating. For all bioassays, naïve 2–4- d old females were used. Females were individually isolated in small vials 1 h before bioassays and then transferred to the bioassay room for acclimation.

## Plant growing

Seeds of broad bean plants (*V. faba* cv. Superaguadulce) were immersed for 24 h in a slurry of water and soil (1:4) to favor root nodulation. Then, seeds were individually planted in plastic pots (9 × 9 × 13 cm) filled with a mixture of agriperlite (Superlite; Gyproc Saint-Gobain PPC Italia, Milan, Italy), vermiculite (Silver; Gyproc Saint-Gobain PPC Italia, Milan, Italy) and sand (1:1:1) and grown in a climate controlled chamber (24 ± 2 °C, 45 ± 10% RH, 12 h:12 h L:D). Plants were watered daily and, from one week post-germination, fertilized with an aqueous solution (1.4 g/l) of fertilizer (5-15-45, N-P-K, Plantfol, Valagro, Italy). Fifteen days old plants with approximately four fully expanded leaves were used for the experiments. Adaxial and abaxial leaf surface of *Vicia faba* plants were considered for behavioural experiments (Fig. S1).

## Plant treatments

### Role of the herbivore walking activity in eliciting induction of plant volatiles

To verify the possible volatile induction in *V. faba* plants by *N. viridula* walking activity on the abaxial leaf surface, behavioral assays were carried out at different time intervals elapsing after the end of the treatments (0 h, 24 h, and 48 h). To obtain plants treated with only footprints (tarsal pressure and tarsal chemical secretions), preventing bug feeding, gravid females with excised stylets were used. For stylet excision, females were previously anaesthetized at −4 °C for 3 min inside a glass tube, in order to immobilize their labium. Afterwards, the stylets were drawn from the labium with an entomological pin (no. 000), to amputate more than half their length using precision micro-scissors under a stereomicroscope (Zeiss Stemi SV8) with optical fiber illumination (Intralux 5000). The treated females were then placed inside a plastic dish (12 cm diameter) allowing them to recover, and used after 1 h to infest plants. A *N. viridula* female with excised stylets was placed for 24 h on the abaxial surface of a leaf located at the second foliar stage level. The insect was set inside a small cage, made from two Petri dishes (35 mm diameter, 10 mm height) with the bottoms substituted with a fine nylon mesh and each rim of the opposite side covered with a small foam rubber ring, which was kept tightened to the leaf with the help of a clip. These treated plants were maintained at controlled conditions (24 ± 2 °C, 45 ± 10% RH, 12 h:12 h L:D) for 24 h and then used for olfactometer bioassays.

To understand whether the adaxial *V. faba* leaf surface plays a different role in the volatile induction associated to herbivore walking activity, *N. viridula* females with excised stylets were individually placed for 24 h on the adaxial leaf surface as described above. In

addition, to exclude the possible role due to the walking position of the bug females and thus the different amount of chemical tarsal secretions due to the insect pressure on the leaf surface, and to exclude the effect of the different light and temperature conditions on the two leaf surfaces, plants were treated with footprints left on the abaxial leaf surface but turning the leaf upside-down. For both treatments, plants were used for olfactometer bioassays 24 h after the end of the treatment.

To evaluate the role of bug gender in the induction of volatiles triggered by walking activity on the abaxial leaf surface and the specificity of these volatiles, *V. faba* plants were exposed respectively to the footprints left by mated *N. viridula* males and by *Murgantia histrionica* females, both with excised stylets. *Murgantia histrionica* was used because it is not a host of *T. basalis* (*Peri et al., 2013*). Considering the difference in weight between *N. viridula* ($0.21 \pm 0.01$ gr) and *M. histrionica* ($0.087 \pm 0.001$ gr) females, two bugs were used inside a clip cage during each treatment. The treatments were implemented following the protocol already described and the plants used for the bioassays 24 h after the end of the insect exposure. Healthy plants were used always as control. As for treated plants, an empty clip cage was applied on the leaf of the second foliar stage for 24 h and the plant was used 24 h after the clip removal. In the case of the treatment represented by footprints left on abaxial upside-down turned leaf, the control was obtained as described above but turning the leaf upside-down.

### Role of feeding and oviposition localization in plant volatile induction

To elucidate whether the abaxial and adaxial *V. faba* leaf surfaces, damaged by feeding or by oviposition (oviposition always combined with feeding), play a different role in volatile induction according to stress localization, *N. viridula* females were individually placed for 24 h on the abaxial or on the adaxial leaf surface as described above. Plants with feeding punctures plus footprints and plants with a combination of feeding punctures, a deposited egg mass and footprints were used for olfactometer bioassays 24 h after the end of the treatment. Healthy plants with an empty clip cage applied on the leaf of the second foliar stage were used as control.

### Behavioural assays

Wasp responses to volatile chemicals from *V. faba* plants subjected to different treatments were investigated with a dual choice Y-tube olfactometer as described by *Moujahed et al. (2014)*. The different treatments were randomly assigned to each olfactometer arm at the beginning of the bioassays and were reversed after testing about 10 parasitoid females. At every switch, the polycarbonate olfactometer was cleaned with water and detergent and the glass parts were changed with cleaned ones. At the end of the bioassays, the glass parts were then cleaned with acetone and baked overnight at 180 °C.

Wasp females were singly introduced into the Y-tube olfactometer at the entrance of the stem and allowed to move freely for 10 min. Their behavior was recorded using a monochrome CCD video camera (Sony SSC M370 CE) fitted with a 12.5–75 mm/F 1.8 zoom lens. The camera lens was covered with an infrared pass filter (Kodak Wratten filter 87 Å) to remove visible wavelengths. Analog video signals from the camera were digitized

by a video frame grabber (Canopus®ADVC 110; Grass Valley, CA, USA). Digitized data were processed by XBug, a video tracking and motion analysis software (*Colazza et al., 1999*). Wasp response was measured in terms of residence time, i.e., the time spent by the wasps in each arm during the entire bioassay.

The Y-tube olfactometer bioassays were carried out as paired choices in which the parasitoids were offered: (1) healthy plant *versus* plant with *N. viridula* female walking activity on abaxial leaf surface at 0 h (4 couples of plants assayed with 39 parasitoids), 24 h (4 couples of plants assayed with 41 parasitoids) and 48 h (4 couples of plants assayed with 41 parasitoids) from the end of the treatment; (2) healthy plant *versus* plant with *N. viridula* female walking activity on adaxial leaf surface (4 couples of plants assayed with 40 parasitoids); (3) healthy plant with upside-down abaxial leaf *versus* plant with *N. viridula* female walking activity on upside-down abaxial leaf (5 couples of plants assayed with 40 parasitoids); (4) healthy plant *versus* plant with *N. viridula* walking activity of male on abaxial leaf surface (4 couples of plants assayed with 34 parasitoids); (5) healthy plant *versus* plant with *M. histrionica* females walking activity on abaxial leaf surface (4 couples of plants assayed with 39 parasitoids); (6) healthy plant *versus* plant with *N. viridula* female walking activity associated with feeding punctures on abaxial (5 couples of plants assayed with 44 parasitoids) or on adaxial leaf surface (4 couples of plants assayed with 34 parasitoids); (7) healthy plant *versus* plant with *N. viridula* female walking activity associated with feeding punctures and oviposition on abaxial (6 couples of plants assayed with 44 parasitoids) or on adaxial leaf surface (6 couples of plants assayed with 58 parasitoids).

All bioassays were conducted from ∼09:00 h to 13:00 h under controlled conditions (26 ± 1 °C, 50 ± 5% RH).

## Collection and analysis of VOCs

Cylindrical glass chambers (inner Ø= 10 cm, $h$ = 30 cm), with a two semi-circular-part joint base made by Teflon with a 2-cm hole in the center to permit the insertion of the plant at the level of collar were used to collect headspace volatiles only from the epigeous part of the plant. Teflon taping was placed around the plant collar to make it tight on the Teflon base. Before each collection, the glass chamber and Teflon base were washed with water and detergent, rinsed with acetone, and baked overnight at 120 °C. A whole plant was inserted in the chamber and flushed with active charcoal filtered air at flux rate of 300 ml min$^{-1}$ for 3 h using a pump NMP 830 KNDC 12V (KNF, Milano, Italy).

VOC emissions by plants were collected using adsorbent traps, placed at the chamber outlet, made by glass tubes filled with PorapakQ (SigmaAldrich; 60 mg, 80–100 mesh), which were pre-cleaned with hexane and then heat conditioned for at least 2 h in a stream of nitrogen (100 ml/min) at 130 °C. After 3 h traps were eluted with 700 µl of hexane, and the resulting extracts were stored at −20 °C in glass vials with Teflon cap liners until used for gas chromatography (GC) analysis. On the basis of behavioral results only the treatments significantly affecting the parasitoid response were used for volatile collection. In particular, VOCs were collected from healthy plants as control, plants with *N. viridula* female walking activity on abaxial leaf surface at 24 h, plants with *N. viridula* female walking activity on adaxial leaf surface; plants with *N. viridula* female walking activity + feeding punctures

+ oviposition on abaxial leaf surface and plants with *N. viridula* female walking activity + feeding punctures + oviposition on adaxial leaf surface. For each treatment, 6 plants were sampled. Blank measurements were carried out before every set of measurements, by sampling air from the chamber, in this case the Teflon base was closed using Teflon.

Gas chromatography-mass spectrometry (GC-MS) analysis was performed on a Hewlett-Packard 5890 GC system interfaced with an HP5973 quadruple mass spectrometer. For each sample, 1 µl of extract was injected into a HP5-MS column (5% diphenyl–95% dimethyl polysiloxane 30 m × 0.2 mm, 0.25-µm film; J & W Scientific, Folsom CA, USA) in splitless mode. Injector and detector temperatures were 260 °C and 280 °C respectively. Helium was used as the carrier gas. The GC oven temperature program was 40 °C for 5 min, then, increased by 10 °C/min to 250 °C. Electron impact ionization spectra were obtained at 70 eV, recording mass spectra from 40 to 550 amu.

Peak area of each detected compound was calculated. Compounds were tentatively identified, based on comparison of RI and mass spectra with those in *Adams (2007)*, http://www.pherobase.com and the NIST 1998 libraries. For (*E*)-2-Hexenal, (*Z*)-3-Hexenyl acetate, Benzaldehyde, $\beta$-Ocimene, $\alpha$-Pinene, $\alpha$-Myrcene, Linalool, $\beta$-Caryophyllene, Octanal, Nonanal, Decanal, Octan-1-ol, Tetradecane and Isomenthone, tentative identifications by GC-MS were confirmed by injection of authentic standards. Standars used were obtained from Sigma-Aldrich (Munich, Germany).

### Statistical analysis

Data were analyzed by linear mixed model (LMM) with the plant treatment as fixed effect and parasitoid nested within each plant pairs as random effects to account for pseudoreplication. Significance of the fixed term in the model was determined using likelihood ratio tests (LRTs) comparing the model with and without the factor in question (*Crawley, 2007*). Because this approach compares models with different fixed effect structures, Maximum Likelihood (ML) was specified in the models instead of Restricted Maximum Likelihood (REML) (*Crawley, 2007*). Model fit was assessed with residual plots. All statistical analyses were carried out with R software, version 3.1.3 (*R Core Team, 2015*).

Data from analysis of volatiles were analyzed by multivariate analysis using projection to latent structures discriminant analysis (PLS-DA) with SIMCA-P+12.0 software program (Umetrics AB, Umeå, Sweden). The projection method determines if samples belonging to the different treatment groups can be separated on the basis of quantitative and qualitative differences in their volatile blends.

## RESULTS

### Role of the herbivore walking activity in eliciting induction of plant volatiles

*Trissolcus basalis* females were able to discriminate between *V. faba* healthy plants and plants with *N. viridula* walking activity on abaxial leaf surface but not with *N. viridula* walking activity on adaxial leaf surface.

In particular, *T. basalis* females significantly preferred volatiles emitted by healthy plants as control compared to plants with *N. viridula* walking activity on abaxial leaf surface,

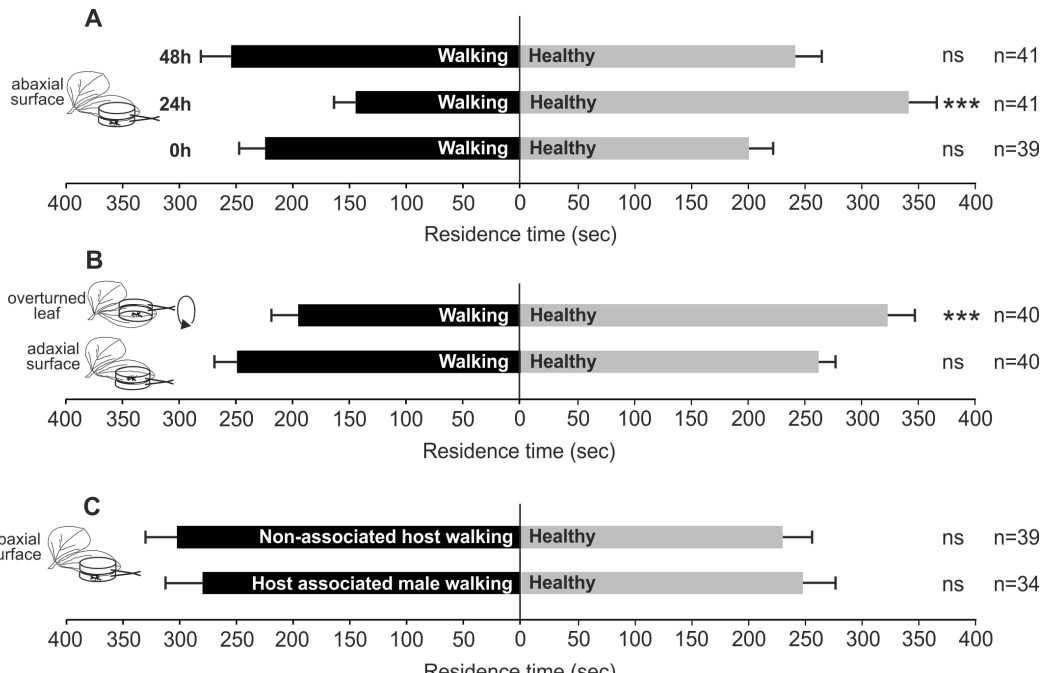

**Figure 1 Possible role of bug walking activity in plant volatile induction.** Response of *T. basalis* females in a Y-tube olfactometer to volatiles from *V. faba* plants treated: (A) with *N. viridula* female walking activity on the abaxial leaf surface, and assayed 0 h, 24 h, and 48 h after the treatment *versus* healthy plants; (B) with *N. viridula* female walking activity on overturned leaf *versus* healthy plants with overturned leaf and plants with *N. viridula* female walking activity on adaxial leaf surface *versus* healthy plants; (C) with *M. histrionica* (non-associated host) walking activity of 2 females on abaxial leaf surface *versus* healthy plants and treated with *N. viridula* walking activity of male on abaxial leaf surface *versus* healthy plants. Bars represent mean (±SEM) of the time spent by wasp females in each arm over an observation period of 600 s. Asterisks (***) indicate $p < 0.001$ by linear mixed model LMM. ns, not significant; n, number of replicates.

assayed 24 h after the end of the treatment ($\chi^2 = 19.016$; $df = 1$; $P < 0.0001$). Whereas differences were not significant when 0 h ($\chi^2 = 0.645$; $df = 1$; $N = 0.422$) and 48 h ($\chi^2 = 0.117$; $df = 1$; $N = 0.733$) were considered (Fig. 1A).

Wasps showed a significant preference for volatiles emitted by the control compared to those released by plants with *N. viridula* walking activity on abaxial leaf turned upside-down ($\chi^2 = 11.162$; $df = 1$; $N = 0.0008$) (Fig. 1B). When comparing plants with *N. viridula* walking activity on adaxial leaf surface and healthy plants, no significant differences were displayed between test and control ($\chi^2 = 0.260$; $df = 1$; $N = 0.611$) (Fig. 1B).

The walking activity of *N. viridula* male ($\chi^2 = 0.56$; $df = 1$; $N = 0.454$) or that of *M. histrionica* females on abaxial leaf surface ($\chi^2 = 2.108$; $df = 1$; $N = 0.147$) did not stimulate a significant response from wasps compared to healthy plants (Fig. 1C).

### Role of feeding and oviposition localization in plant volatile induction

*Nezara viridula* oviposition, combined with feeding and footprints, on plant surface induces OIPVs triggering the parasitoid response only when egg deposition occurs on abaxial leaf surface. In particular, parasitoid females showed a significant preference for volatiles

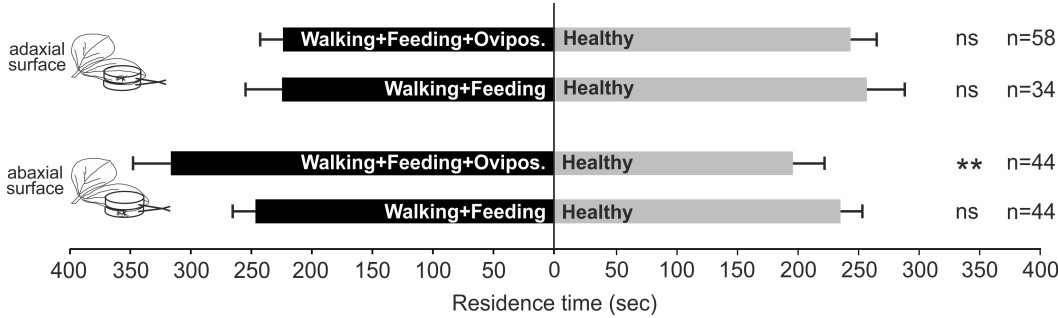

**Figure 2** **Role of stress localization in the induction of plant volatiles.** Response of *T. basalis* females in a Y-tube olfactometer to volatiles emitted by (1) *V. faba* plants with *N. viridula* female walking activity associated with feeding punctures and oviposition on the adaxial leaf surface *versus* healthy plants; (2) plants with *N. viridula* female walking activity associated with feeding punctures on the adaxial leaf surface *versus* healthy plants; (3) plants with *N. viridula* female walking activity associated with feeding punctures and oviposition on the abaxial leaf surface *versus* healthy plants and (4) plants with *N. viridula* female walking activity associated with feeding punctures left on the abaxial leaf surface *versus* healthy plants. Bars represent mean (±SEM) of the time spent by wasp females in each arm over an observation period of 600 s. Asterisks (**) indicate $p < 0.01$ by linear mixed model LMM. ns, not significant; n, number of replicates.

released by plants exposed to *N. viridula* walking, feeding and oviposition activities on abaxial leaf surface ($\chi^2 = 7.331$; $df = 1$; $N = 0.007$) compared to the control (Fig. 2), whilst, when the treatment was located on adaxial leaf surface no significant choice was displayed ($\chi^2 = 0.247$; $df = 1$; $N = 0.620$) (Fig. 2). In the case of *V. faba* plants with *N. viridula* walking and feeding activities on abaxial ($\chi^2 = 0.164$; $df = 1$; $N = 0.685$) or adaxial leaf surface ($\chi^2 = 0.392$; $df = 1$; $N = 0.531$), no preference for test and control was shown (Fig. 2).

*Plant VOC analysis*

The following 23 compounds were detected in *V. faba* plants differently treated: (*E*)-2-Hexenyl butyrate, Isomenthone, Hexyl butyrate, Undecan-2-one, $\alpha$-Myrcene, unknown 1, Ethylbenzene, $\beta$-Caryophyllene, $\alpha$-Pinene, Hexan-1-ol, unknown 2, Octanal, Benzaldehyde, (*Z*)-3-hexen-1-ol, (*E*)-2-Hexenal, Tetradecane, (*E*)- $\beta$ Ocimene, (*Z*)-3-hexenyl acetate, Octan-1-ol, Decanal, Linalool, 6-Methyl-5-hepten-2-one, nonanal, 1-Octen-3-ol. The PLS-DA comparison including samples of all treatments, resulted in a model with one significant principal component (PC1; R2X $= 0.175$; R2Y $= 0.148$; Q2 $= 0.055$; Fig. 3). The model separated healthy plants and plants with *N. viridula* walking on adaxial leaf surface from plants with *N. viridula* walking on abaxial leaf surface and plants with *N. viridula* walking associated with feeding punctures and oviposition on abaxial or adaxial leaf surface. In particular, the volatile blend emitted by plants with footprints left on abaxial leaf surface is different from that emitted by plants with footprints left on adaxial leaf surface and from healthy plants. Examination of the loading plot showed that a group of 8 compounds contributed the most to explaining the variation in the model (Fig. 3B). These compounds have the following corresponding VIP values (variable importance for the projection): (*E*)-2-Hexenyl butyrate $= 1.54$; Isomenthone $= 1.48$; Hexyl butyrate $= 1.31$; Undecan-2-one $= 1.28$; $\alpha$ Myrcene $= 1.21$; Unknown $= 1.10$; Ethylbenzene $= 1.10$; $\beta$-Caryophyllene $= 1.05$.

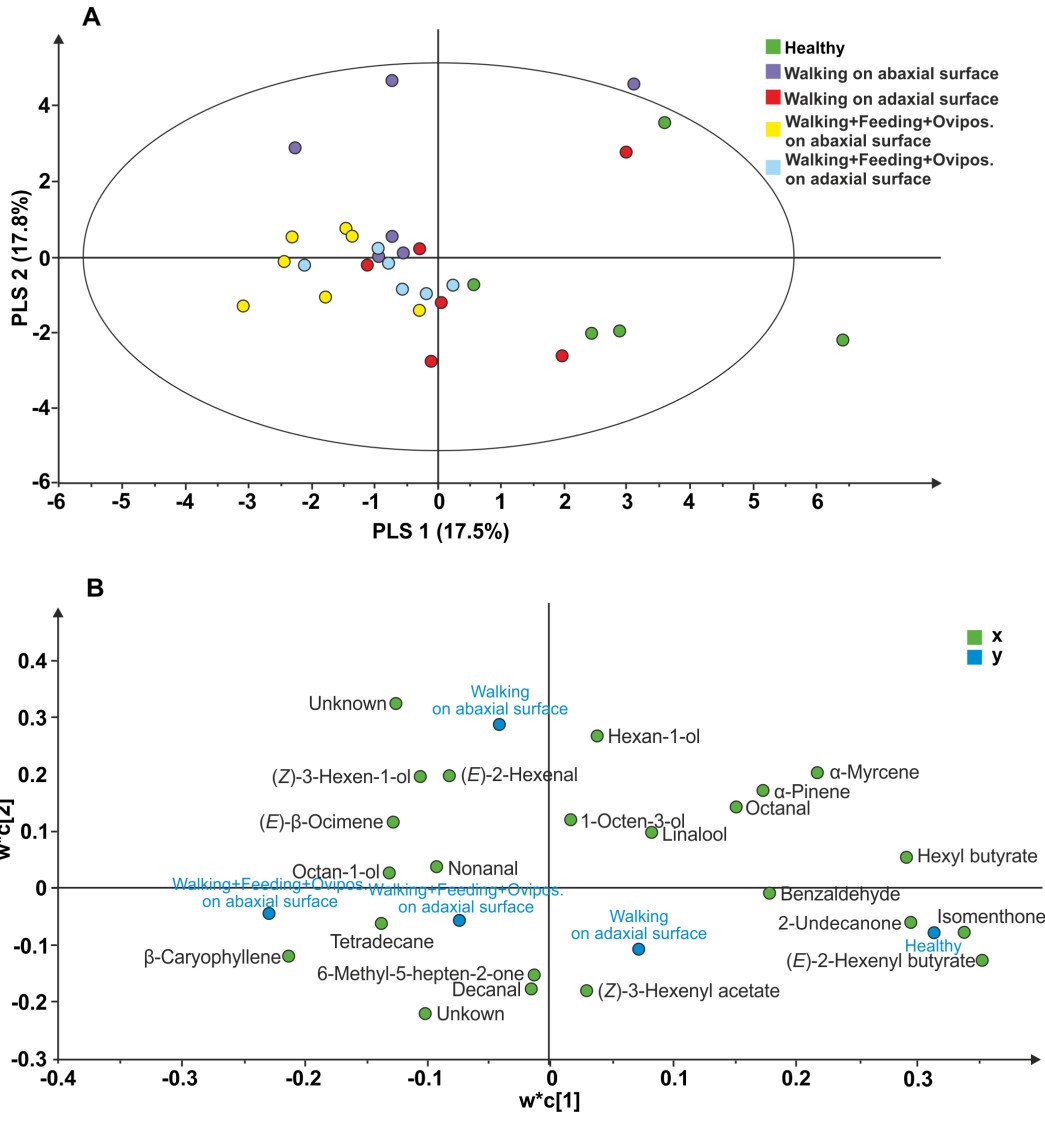

**Figure 3** **Projection to latent structures discriminant analysis (PLS-DA) comparison of the volatile compounds emitted by individual *V. faba* plants.** (A) Score plot of the samples, with the percentage of explained variation in parentheses. The PLS-DA resulted in a model with one significant principal components (PCs). The ellipse defines the Hotelling'sT2 confidence region (95%). (B) Loading plot of the first two components of the PLS-DA, showing the contribution of each of the compounds toward the model.

## DISCUSSION

Our study suggests that the herbivore walking activity on the substrate elicits plant volatile induction. The egg parasitoid *T. basalis* was able to discriminate between volatiles from healthy plants and volatiles from plants with footprints of *N. viridula* females. However, the parasitoid response was time interval dependent, as it was recorded after 24 h from the treatment, while no response was shown at the time intervals of 0 and 48 h after the end of bug walking activity. The leaf surface is actively involved in this induction. In fact, only when the herbivore walking activity occurred on the abaxial leaf surface the parasitoid

response was shown. This is demonstrated by the fact that turning the leaf upside-down and treating the abaxial leaf surface the parasitoid response did not change. In addition, there were no behavioral responses by the parasitoid when leaf treatment occurred with walking activity of *N. viridula* males and of the non-associated host *M. histrionica* females. The combination of walking and feeding activities on adaxial and abaxial leaf surface did not induce wasp behavioral responses. Furthermore, our research reveals that *N. viridula* oviposition (combined with walking and feeding) elicited the induction of OIPVs when it has occurred on abaxial leaf surface. This result confirms the induction of OIPVs in *V. faba* by *N. viridula*, already reported in previous papers where the oviposition was generally recorded on the abaxial leaf surface, as happens in natural conditions, because the plant exposition was carried out in a wood-framed, nylon mesh cage (*Colazza et al., 2004*) or in a net bag (*Moujahed et al., 2014*) with *N. viridula* females free to explore the whole plant.

It is possible to hypothesize that, in our system, the walking activity of *N. viridula* female on abaxial *V. faba* leaf surface elicited the emission of putative induced plant volatiles. The behavioural data seem to be supported by volatile analyses since the PLS-DA showed that the whole blend of plant volatiles changes according to stress localization, i.e., when footprints are left on the abaxial or adaxial leaf surface. An effect on wasp behaviour of volatilization of precursors deposited by the walking insects on leaf surface could be excluded considering that the wasp response was recorded only when the abaxial leaf surface, but not the adaxial leaf surface, is contaminated. Although the induced volatiles associated with the herbivore walking activity did not attract the parasitoid but favoured a repellence effect, they actually might act similarly to a synomone. In fact, our hypothesis is that these induced volatiles could give the parasitoid information that the plant has been visited by the herbivore but it is not there anymore. This repellence for 'old' (24 h) footprint is not showed for fresh footprints (0 h) as they could indicate the possibility of herbivores around. Considering that the parasitoids are under selection pressure to maximize their foraging efficiency in order to improve their ecological fitness (*Tamiru, Zeyaur & Bruce, 2015*), it is fundamental that they should not waste time exploiting these chemical cues, which may be unreliable indicators of the egg presence. The emission of putative induced volatiles associated with herbivore walking activity is not due to the simple contact with the herbivorous insect but probably to the secretions released at tarsal level and probably produced by tibial glands (R Romani, pers. comm, 2016).  These secretions, which concur to the herbivore adhesion on the leaf surface, interact with the plant in a specific manner depending on whether the interaction occurs with the abaxial or the adaxial leaf surface. In addition, the adhesion of the *N. viridula* tarsi on the substrate, and in particular of the claws during the stationary contact on the surface, does not provoke evident mechanical damage on both leaf surfaces as morphological investigations by scanning electron microscope reveal no differences between healthy and footprint-treated *V. faba* leaf surfaces (R Romani, pers. comm., 2016). Our results clearly show that the two leaf surfaces interact differently with *N. viridula* walking activity. This phenomenon could be explained considering the following morphological, anatomical and structural differences between abaxial and adaxial leaf surfaces. The wax composition of the abaxial leaf surface may differ from that of the adaxial surface (*Müller & Hilker, 2001*); leaf palisade

mesophyll cells are beneath the adaxial leaf surface and spongy mesophyll in the lower half; stomatal density is higher on the abaxial surface respect to adaxial surface of leaves (*Willmer & Fricker, 1996a*); abaxial guard cells are typically larger and stomatal pores are wider under conditions favouring opening (*Willmer & Fricker, 1996b*); finally, gas exchange between a leaf or leaflet and the atmosphere occurs mainly via abaxial stomata (*Lu, 1988*). In the case of *V. faba*, leaves have stomata on both abaxial and adaxial epidermis, but the number of stomata per mm$^2$, the stomata width and length on the abaxial epidermis of the leaflet is higher than on the adaxial epidermis (*Pekşen, Pekşen & Artik, 2006*). The pad secretions of *N. viridula* are released during walking on the substrate (G Salerno, pers. comm., 2016) but how they interact with leaf surface or if /how they move into the leaf remains unknown. Moreover, the exposure to light and/or temperature of the tarsal secretions could vary from abaxial and adaxial leaf surface, but in our bioassays we can exclude the effect of the different light and temperature conditions on the two leaf surfaces since no differences were recorded in the *T. basalis* responses towards plants with the abaxial leaf surface with *N. viridula* walking and plants with the upside-down turned leaf.

The two leaf surfaces interact differently not only with *N. viridula* walking but also with the eggs laid by the herbivore. It is known that plants react to herbivore oviposition activating defensive responses. In particular egg deposition, also associated with wounding, induces the emission of volatiles attracting antagonists of the herbivores. The majority of herbivore insects oviposit on plant leaves (*Hilker & Meiners, 2011*) preferring the abaxial leaf surface rather than the adaxial (*Müller & Hilker, 2001*). Oviposition on adaxial leaf surface exposes eggs to sunlight, to unfavourable microclimatic conditions (*Willmer, 1986*) and to egg predators and parasitoids (*Müller & Hilker, 2001*). As recorded for other stink bug species (*Tood & Herzog, 1980*), *N. viridula* tends to oviposit on the abaxial leaf surface (*Colazza & Bin, 1995*). Our data show that *N. viridula* oviposition (combined with feeding and footprints) on plant surface induces OIPVs triggering the parasitoid response only when egg deposition occurs on abaxial leaf surface. However, these results are not explained by the PLS-DA, as the model does not separate between volatiles emitted in response to oviposition occurring on the abaxial or adaxial leaf surfaces. However, in our analysis $(E)$–$\beta$-caryophyllene appears to be a VIP compound confirming its potential role as synomone for *T. basalis* as previously hypothesized by *Colazza, McElfresh & Millar (2004)*.

PLS-DA analyses take into account all volatiles but the wasps likely use a specific subset of compounds of the total blend (*Clavijo McCormick, Unsicker & Gershenzon, 2012*). Thus, a mismatch between behavioral responses and chemical analyses could be due to the fact that the parasitoids focused on some key volatiles associated with *N. viridula*-egg deposition on the abaxial leaf surface which constitute the active blend, whereas PLS-DA takes the whole blend into account.

In conclusion, our results confirm that the signals mediating the interaction between plants, herbivores and parasitoids are effective and finely tuned since they guarantee maximization of the chance to find the suitable host. In particular, data reported in this paper elucidate, first, the role of the herbivore walking activity in a simplified experimental set up (walking alone, without feeding and oviposition, only occasionally occur in nature),

and second, the key role of the abaxial leaf surface in mediating the volatile communication between *T. basalis* and its host.

## ACKNOWLEDGEMENTS

We are grateful to Andrea Luchetti for rearing the insects and to Daniela Fortini and Cesare Dentini for growing *Vicia faba* plants. We thank Anna Laureti for helping in data collection.

### Funding

Funding was provided by ''Fondo d'Ateneo per la Ricerca di Base 2014'' of Perugia University and by the project MIUR –PRIN 2010-2011 ''Going to the root of plant productivity: how the rhizosphere interacts with the aboveground armament for indirect and direct defense against abiotic and biotic stressors (PRO-ROOT)'', and by the Marie Skłodowska-Curie Research and Innovation Staff Exchange (RISE) H2020-MSCA-RISE-2015 of the European Union as part of the project ''Impact of invasive alien true bug species in native trophic webs'' –INVASIoN (GA 690952). The funders had no role in study design, data collection and analysis, decision to publish, or preparation of the manuscript.

### Grant Disclosures

The following grant information was disclosed by the authors:
Fondo d'Ateneo per la Ricerca di Base 2014.
Research and Innovation Staff Exchange (RISE): H2020-MSCA-RISE-2015.
INVASIoN: GA 690952.

### Competing Interests

The authors declare there are no competing interests.

### Author Contributions

- Francesca Frati conceived and designed the experiments, performed the experiments, wrote the paper, reviewed drafts of the paper.
- Antonino Cusumano analyzed the data, reviewed drafts of the paper.
- Eric Conti, Stefano Colazza and Ezio Peri conceived and designed the experiments, reviewed drafts of the paper.
- Salvatore Guarino and Letizia Martorana performed the experiments.
- Roberto Romani conceived and designed the experiments, reviewed drafts of the paper, he provided SEM picture included as supplemental figure.
- Gianandrea Salerno conceived and designed the experiments, performed the experiments, contributed reagents/materials/analysis tools, prepared figures and/or tables, reviewed drafts of the paper.

## Supplemental Information

Supplemental information for this article can be found online at http://dx.doi.org/10.7717/peerj.3326#supplemental-information.

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
