# Peer review of "Foraging behaviour of an egg parasitoid exploiting plant volatiles induced by pentatomids: the role of adaxial and abaxial leaf surfaces"

_PeerJ, doi:10.7717/peerj.3326_

## Round 0.1 · original submission · Minor Revisions

Please pay particular attention to the comments raised by Rev1.

Reviewer 1 ·

Basic reporting

English is clear but needs some polishing by a native speaker/ editor.
Enough background information for the reader is given in the introduction. There could be more recent references cited e.g. on egg-induced defences.
I would merge figure 1-3 to one figure on the effect of foot prints (and put a-c), which makes a comparison easier. I would also provide a figure that compares a picture of the adaxial with the abaxial leaf side? I assume that they visually differ?
I would provide the number of replicates/responding wasps next to each bar or at least mention them in the legend.

Experimental design

Research questions are in my view not specific enough to understand what exactly has been tested. I would start with the general research question and then state more specific, that behavioral tests were conducted to test…and these were followed by chemical analysis to confirm/ not confirm the behavioral observations. The knowledge gap is stated but could be clearer, i.e. line 109-110: a second phase of this study…I don’t understand what is meant with the second phase.
Methods were generally described with sufficient detail/ information to replicate.
Plant volatile analysis to me is not completed. I miss a (tentative) identification of compounds by comparison of the mass spectra with mass spectral databases like NIST libraries. Information on the mass spectra alone is to me not meaningful and makes comparison to previous chemical analysis in the same system (Colazza et al. 2004) more difficult.
Statistical analysis of two-choice bioassays is performed with a complicated model instead of simple t-tests. I don't really see the reasoning here behind as only the residence times within the tested treatments are compared and not between the treatments (See e.g. Cusumano et al. 2015).
Line 134: Were plants only 5 days old?

Validity of the findings

I would recommend to provide compound identifications and reconsider statistical analysis of the behavioral data and/or provide comparison between different treatments.
Line 376-378 is not clearly mentioned in the results.
I am a bit puzzled why the wasps are repelled by volatiles of plants with footprints 24h after treatment. Why then not also by volatiles from plants with footprints and feeding 24h after treatment? Therefore, the argumentation in line 336-338 to me does not really help. And if the reason why the wasps are repelled to 24h old footprints is that they are ‘old’ and indicate that the herbivore is not there anymore, than the fresh footprints (0h) should be attractive as they would indicate herbivores around? Are fresh feeding and footprints of females more reliable indicators for egg parasitoids? Has this combination been tested at 0h as well (maybe in another study)? Clearly, OIPVs are most reliable but no differences were seen in the blends. Studies on OIPVs in Brassicaceae report similar problems on separating OIPVs from healthy plants (e.g. Ponzio et al. 2016; Fatouros et al. 2012). Its seems that only minor shifts in ratios of some compounds take place not picked up by a PCA. It would help if differences in identified compounds would be analyzed by other statistical means. In the previous study of Colazza et al. 2004, beta-caryophyllene was shown to play a role. It would be interesting to see whether this is the case in the present study too.

Reviewer 2 ·

Basic reporting

This study is a welcome addition to area of the chemical communication of parasitoid, its host insect and host plant, providing valuable information on the tritrophic relationship . The study was well-designed, the manuscript is well-written with clear points made for each claim. However, including more data on the chemical analysis will make the authors' claims more clear.

Experimental design

The experiments were well designed to obtain clear answers to the key questions in this study. The method section provides enough details so that other researchers can understand and replicate the study.

Validity of the findings

The findings and authors' claims are well supported by the results obtained and the data presented. However, providing more data on the chemical analysis is desired.

Additional comments

This paper, describing the responses of host plants to walking activities of stink bugs and subsequent behavioral responses of egg parasitoids, is a good example of fine scientific research, providing valuable information to the existing knowledge on the tritrophic interaction among them. The new finding showing walking activity of stink bug on abaxial surface of plant leaves elicits significant and distinct responses from the plant and egg parasitoid opens a new fine-tuned research area in insect-plant relationship and the foraging behavior of egg parasitoids. Although the analysis and comparison on the profiles of volatile chemicals indicates differences between treatments, however, it would be good to show the names and contents of the volatile compounds identified so that readers can better understand the outcome of this study. The English writing is good.

Minor comments
1. A space should be provided between number and unit (currently inconsistent throughout the manuscript).

2. L60
"Please add "it" after "in addition".

3. L94
Please consider revising this sentence. It is a bit confusing what "short-range induced synomones".

4. L101
What does "atmospheric agetnts" mean? It can be more specific.

5. L187
Please correct the use of parenthesis for the reference.

6. L296
Please list all of the 22 compounds identified. This will be informative.

---

## Round 0.2 · accepted · Accept

Thank you for your very detailed and clear response to the reviewers.